# Human Oncogenic Epstein–Barr Virus in Water and Human Blood Infection of Communities in Phayao Province, Thailand

Sutida Pongpakdeesakul [1], Tipaya Ekalaksananan [2,3], Chamsai Pientong [2,3], Niti Iamchuen [4], Surachat Buddhisa [5], Khwanruedee Mahingsa [6], Arunee Pingyod [6], Wanwipa Sangsrijun [6], Supaporn Passorn [1], Peechanika Chopjitt [7], Sureewan Duangjit [8] and Sureewan Bumrungthai [3,9,10,*]

[1] Division of Biotechnology, School of Agriculture and Natural Resources, University of Phayao, Phayao 56000, Thailand
[2] Department of Microbiology, Faculty of Medicine, Khon Kaen University, Khon Kaen 40002, Thailand
[3] HPV & EBV and Carcinogenesis Research Group, Khon Kaen University, Khon Kaen 40002, Thailand
[4] School of Information and Communication Technology, University of Phayao, Phayao 56000, Thailand
[5] Department of Medical Technology, Faculty of Allied Health Sciences, Burapha University, Chonburi 20131, Thailand
[6] Thalassemia Unit, University of Phayao Hospital, University of Phayao, Phayao 56000, Thailand
[7] Faculty of Public Health, Chalermphrakiat Sakon Nakhon Campus, Kasetsart University, Sakon Nakhon 47000, Thailand
[8] Division of Pharmaceutical Chemistry and Technology, Faculty of Pharmaceutical Sciences, Ubon Ratchathani University, Ubon Ratchathani 34190, Thailand
[9] Division of Microbiology and Parasitology, School of Medical Sciences, University of Phayao, Phayao 56000, Thailand
[10] Division of Biopharmacy, Faculty of Pharmaceutical Sciences, Ubon Ratchathani University, Ubon Ratchathani 34190, Thailand
* Correspondence: sureewan.b@windowslive.com

**Abstract:** Water can contain pathogenic viruses. Many studies on RNA virus sources have shown that water can transmit them. However, there are few reports on pathogenic DNA virus transmission through water, such as adenovirus, which pose a widespread public health risk. Therefore, this study aimed to show waterborne viral transmission by detecting viruses in pooled human whole blood samples, tap water, and natural water from Mueang District, Phayao Province, Thailand, using a metagenomic approach. Viral prevalence in whole blood samples was measured by polymerase chain reaction (PCR) and quantitative PCR (qPCR), and environmental factors that affect viral infection were assessed. Metagenomics results showed that Epstein–Barr virus (EBV) members were among the prominent cancer-associated oncogenic DNA viruses detected in human blood and all water types similar to the EBV reference sequence (NC_007605). There were 59 out of 813 (7.26%) human whole blood samples that were positive for EBV DNA based on PCR and qPCR for the *EBNA-1* and *EBNA-2* genes. Water- and blood-borne human oncogenic EBV should be a concern in tap water treatment and blood transfusion in patients, respectively. Therefore, the detection of EBV in water suggests that transmission via water is possible and should be investigated further.

**Keywords:** EBV; blood; blood-borne virus; tap water; natural water

## 1. Introduction

Rural residents have high disparities, even in access to clean water, which is a fundamental factor in life [1–3]. Poverty, poor edification, low health knowledge, lifestyle, insufficient sanitation, poor nutrition, exercise, and environmental factors (such as clean water) have been identified as primary factors leading to the high infectious disease incidence among women in developing countries [4,5]. Similarly, antibiotic use was larger in rural (23.6%) than in urban (20.2%) areas compared [3]. The International Agency for Research on Cancer (IARC) estimates that 15–20% of cancers are associated with infectious agents [6].

Several DNA and RNA viruses from various cancer-associated viral families are found in water [7]. A leading problem is an increasing trend in oral squamous cell carcinoma (OSCC) in patients aged under 45 years (20–34 years) in the Netherlands between 1989 and 2018, with an annual percentage change of 2.4% (95% confidence interval (CI): 1.1–3.8%) [8,9], associated with oncogenic DNA viruses. Ninety-eight studies found that colorectal, breast, kidney, pancreas, and uterine cancers are increasing in younger age groups; although, the mechanisms remain unclear [10]. This study is concerned with the quality of water that might be a source of carcinogens, such as viruses, which can cause human disease and many cancers.

Plant, human, and animal viruses can be transmitted through water. Viruses isolated from trees, soil, and water from the abovementioned areas may originate from infected plant debris. Human and animal waste has also been identified as a possible source of enteric viruses in water [11–13]. About 25% of the global population consumes fecally contaminated water, which may contain bacteria, protozoa, and viruses that can cause many diseases in humans [14], and pose a public health risk [15]. Tap-water-borne viruses such as norovirus [16] and torque teno virus (TTV) are of particular concern since many cause disease [17]. Human adenovirus, rotavirus, and hepatitis A virus (HAV) have been found in treated wastewater treatment [15,18]. Adenovirus, astrovirus, norovirus, hepatitis E virus (HEV), HAV, coronavirus, poliovirus, coxsackievirus, echovirus, and rotavirus have been found in drinking water [19]. The conventional methods used to detect waterborne viruses include polymerase chain reaction (PCR), quantitative PCR (qPCR), and nested PCR [15].

Twenty-first-century aquatic virome studies were first performed using shotgun metagenomics sequencing [11,12]. HEV and 26 different viral families were detected in raw and tap water by next-generation sequencing [20,21]. Shotgun sequencing has also been used to detect blood-borne viruses, including uncultured DNA viruses such as human immunodeficiency virus (HIV), hepatitis B virus (HBV), hepatitis C virus (HCV), human T cell lymphotropic virus (HTLV), West Nile virus, human gammaherpesvirus 8 (HHV8), TTV [22], human Gemykibivirus-2 [23], paleoviruses, human papillomavirus (HPV) type 27, Merkel cell polyomavirus, astrovirus MLB2, human pegivirus, cytomegalovirus, parvovirus B19, and HEV. Families *Anelloviridae*, *Flaviviridae*, *Picornaviridae*, and *Herpesviridae* have also been detected [24]. There are reports on the risk of raw and tap water being contaminated with water-borne viruses, including RNA virus families such as *Picornaviridae*, *Caliciviridae*, *Hepeviridae*, *Reoviridae*, and *Astroviridae* and the DNA virus family *Adenoviridae*, which causes illnesses with various symptoms from mild to severe gastroenteritis to meningitis, respiratory disease, conjunctivitis, myocarditis, paralysis, and hepatitis [25]. Several DNA and RNA viruses belonging to various viral families are associated with human cancers. To date, seven viruses have been classified by the IARC as "carcinogenic to humans" (group 1): HBV, HCV, Epstein–Barr virus (EBV), Kaposi's sarcoma herpes virus, also known as human herpesvirus 8 (HHV-8), HIV type-1, HTLV type-1, and HPV (several genotypes) [7]. Long-term consumption of polluted water with 3-methylcholanthrene can induce malignant cell transformation [26]. Moreover, the sources of oncogenic DNA viruses, such as members of the *Herpesviridae* family (e.g., EBV or *Human gammaherpesvirus 4*), which cause oropharyngeal cancer, Burkitt lymphoma, Hodgkin's lymphoma [27–29], nasopharyngeal carcinoma (NPC) [30,31], and posttransplant lymphoproliferative diseases have not been identified in water before [32].

The oral route is the primary route of EBV transmission. EBV is known to be transmitted via saliva but rarely spreads through semen, blood, or organ transplantation [23,24]. EBV can be spread by using objects, such as a toothbrush, drinking glass, or eating utensils, which an infected person recently used or by sharing their food [32]. Nevertheless, organ transplantation and blood transfusion can cause EBV spread, and blood transfusion recipients should be considered at particularly high risk [32]. Half of all mothers and one-week-old newborns were EBV positive based on nested PCR and 60.6% based on immunoglobulin G (IgG) [33,34]. EBV infection was not observed before 3 months of age but slowly emerged thereafter, reaching a cumulative rate of 1.7%, 11.6%, and 21.5% at

6, 12, and 14 months, respectively [35]. The transmission routes leading to EBV infection between three months and adulthood other than human-to-human transmission remain unclear, and it is unknown whether it can be transmitted through environmental sources such as water.

EBV has a double-stranded linear DNA genome containing approximately 100 genes surrounded by a protein capsid. *Latent EBV nuclear antigen (EBNA)* genes include *EBNA-1, EBNA-2, EBNA-3A, EBNA-3B, EBNA-3C, EBNA-LP, LMP-1*, and *LMP-2A/2B* [36,37]. EBV infection and the associated immune response occur mainly during the first years of life but may also occur during adolescence or adulthood [38,39]. EBV-positive antibodies are present in the serum of >90% of children and adults worldwide. Another study in Thailand found that the prevalence of anti-EBV IgG was 34.9–95% [40,41]. However, serum antibody IgG levels were unrelated to the disease [40,42]. The prevalence of EBV infection in Thailand using specific immunoglobulin M (IgM) antibodies was 11–13%, representing acute or primary infection associated with EBV DNA. An analysis of anti-EBV IgM and EBV DNA in 80 serum samples from children aged <2 years found that three (3.8%) were positive for IgM and none were positive for EBV DNA using PCR [40,41].

*EBNA-1* DNA was detected in 72.5% of whole blood samples from Indonesian NPC patients using qPCR [30]. *EBNA-1, EBNA-2*, and *LMP-1* genes were detected in 56% of whole blood samples from Chinese HIV patients tested for EBV DNA using nested PCR [43]. The *LMP-1* gene was detected in 61% of healthy blood donors in Qatar [44]. In addition, the *EBNA-2* gene was detected in 52.6% of Qatari samples tested using qPCR [45]. Moreover, 9.5% of whole blood samples were positive for EBV DNA in diffuse large B cell lymphoma [46]. The prevalence of EBV-positive diffuse large B-cell lymphoma in East Asians (8.7–11.4%) was higher than in Western countries (5%) [32,33]. A study of blood donors in Ouagadougou, Burkina Faso, found a lower EBV level of 5.1% using qPCR [47]. The PCR-based prevalence of EBV in normal oral exfoliated cells (the most commonly found EBV infection) was 19% in Thailand, lower than in other countries [48]. In situ hybridization, Southern blotting, and dot blotting have all been used to diagnose and monitor primary EBV infection. Measuring EBV DNA in blood samples by qPCR is important for monitoring EBV-associated disease [32,49–51]. Molecular techniques such as qPCR can be used to detect EBV in various samples, such as serum, whole blood, unfractionated blood, tissue biopsy, and peripheral blood mononuclear cells (PBMCs) [49]. A good correlation between EBV DNA loads in whole blood and PBMCs was found by qPCR [52,53]. Few studies have investigated EBV viremia in healthy individuals [47].

Environmental factors, including sunlight/vitamin D, smoking, and body mass index (BMI), affect EBV infection [38]. EBV infection is not recognized as a national problem in Thailand due to insufficient knowledge. Information on EBV pathogenesis can be found only in textbooks and among a small group of researchers in Thailand. However, the increasing incidence of many cancers has led to appreciable data directly associating cancer with EBV infection. The presence of EBV in water may be associated with human blood transmission, which poses a widespread public health risk.

Phayao Province, Thailand, is a rural small province with a population of 472,356 in 2019. Mueang District, in the center of Phayao Province. Mae Ka and Wiang Subdistrict in the center of the Mueang District. Wiang Subdistrict residents use tap water from the Kwan Phayao lake (19°10′32.5″ N 99°52′04.0″ E), while those of the Mae Ka Subdistrict (No. 1) use tap water from Huai Na Poi Brook (19°00′59.2″ N 99°53′01.9″ E) and also Kwan Phayao lake. Water from Kwan Phayao is contaminated by sewage from communities, livestock, and hospitals. Nevertheless, the Provincial Waterworks Authority sources tap water from Kwan Phayao. In contrast, tap water sourced from Huai Na Poi, originates from a deep and wide forest basin, with water flowing in from the mountains and is uncontaminated by humans. However, it is relatively cloudy and smells due to humus deposition. Because tap water from Huai Na Poi is directly produced by the villages, it has a lower quality, evident from its turbidity.

Therefore, this study aimed to demonstrate that water is a source of viral transmission by detecting viruses in pooled human whole blood, tap water (Mae Ka and Wiang Subdistricts), and natural water (Huai Na Poi and Kwan Phayao) in the Mueang District, Phayao Province, Thailand, using metagenomics. Tap water from the Mae Ka and Wiang Subdistricts was collected for quality evaluation. The prevalence of oncogenic DNA (EBV) viruses in whole blood samples was determined using PCR and qPCR. Environmental factors that affect viral infection were also evaluated.

## 2. Materials and Methods

### 2.1. Water Sampling Methods

Tap water from Mae Ka Subdistrict (No. 1) and Wiang Subdistrict was collected from Mueang Phayao District, Phayao Province (5 L for chemical analysis and 1 L for microbial analysis). Water quality was tested in duplicate by chemical analysis to detect turbidity, color (Pt.Co), pH, total dissolved solids, total hardness (calcium carbonate [$CaCO_3$]), chloride ($Cl^-$), fluoride ($F^-$), nitrate ($NO^{3-}$), nitrite ($NO^{2-}$), arsenic (As), cadmium (Cd), chromium (Cr), copper (Cu), iron (Fe), lead (Pb), manganese (Mn), mercury (Hg), sulfate ($SO_4^{2-}$), and zinc (Zn). Microbial analysis was performed for *Escherichia coli* (*E. coli*) and total coliform bacteria. The Provincial Waterworks Authority's standard value was used as a reference. We used an in-house testing method (TE-CH-037) based on standard methods for examining water and wastewater: APHA, AWWA WEF, 23rd Edition, 2017; WEF, 23rd Edition, 2017, Part 3030 E, 3120 B, 3125 B, 9221; APHA-AWWA (2005); and APHA-AWWA (2017). *E. coli* and coliforms were detected by Multiple tube fermentation technique [most probable number (MPN)].

Five L each of tap water (Mae Ka Subdistrict (No. 1) and Wiang Subdistrict) and natural water (Huai Na Poi and Kwan Phayao) were collected from Mueang Phayao District, Phayao Province, for water filtration and metagenomic analysis. Tap water was collected by swabbing the faucet with alcohol wipes, turning on the water, letting it run for about 1 min, and then rinsing the container a few times before collecting the sample. Natural water samples were collected from under the surface in sterile glass containers and stored on ice for future metagenomic analysis.

### 2.2. Water Filtration

A total of 5 L of each water sample was filtered using a Rocker 300 vacuum pump (Waller Chemical Thailand Co., Ltd., Bangkok, Thailand) using MF-Millipore pore size 0.025 μm MCE filter nitrocellulose membranes (MF-MilliporeTM, 47 mm; Merck Millipore Ltd., Tullagreen, Carrigtwohill, Co. Cork, Ireland) and on top with 0.45 μm membrane filter. Next, DNA was extracted from all microorganisms, including viruses, on the filter membranes using DNA extraction from filter membranes method. It should be noted that the smallest viruses are about 0.02–0.4 μm in diameter (EBV is approximately 0.12–0.18 μm in diameter) [54–57]. Microfiltration is generally considered appropriate for quantifying microbes and viruses 0.025–10.0 μm [58,59] in water samples [60,61].

### 2.3. Specimens

Eight hundred thirteen human whole blood samples were collected from donors living around the natural water sources in the Mae Ka and Wiang Subdistricts, Phayao Province. Samples were collected from individuals aged 3–90 years, along with information on their sex, health status, and life history, including congenital disease, BMI, family cancer history, exercise, alcohol consumption, smoking, secondhand smoke, drinking water source, cleaning of water for consumption, and type of water used for brushing teeth, serving spoons, and eating fresh fruit and vegetables. The sample size was calculated as $N = Z^2_{1-a} P(1 - P)/d^2$ for an EBV prevalence (P) of about 2% [62]: Z = 1.96 for a 95% confidence level and d = 0.01. This study was approved by the Committee on Human Research Ethics in Health Sciences and Science and Technology, University of Phayao (Mae Ka, Thailand; 1.3/023/63). All procedures involving human participants performed in

the study were in accordance with the ethical standards of the Declaration of Helsinki, the Belmont Report, the Council for International Organizations of Medical Sciences guidelines, and the International Conference on Harmonization in Good Clinical Practice. Informed consent was obtained from all subjects and/or their legal guardian(s).

## 2.4. DNA Extraction

### 2.4.1. DNA Extraction from Filter Membranes

Filter membranes were used for total DNA extraction and metagenomic analysis. Cell lysis used a cell lysis solution 4 mL soak and mix by vortex (10 mM Tris-hydrochloric acid (pH 7.8), 5 mM ethylenediaminetetraacetic acid (EDTA), 0.5% sodium dodecyl sulfate, and 1 mL lysozyme) and proteinase K 0.5 mL (stock 20 mg/mL) incubated at 37 °C and 55 °C for 30 min and 60 min, respectively, and allowed to cool at room temperature (RT) for 20 min. Potassium acetate (5M) 4 mL was added to precipitate the protein, and samples were centrifuged at 12,000× $g$ at 4 °C for 10 min. DNA was purified using (1:1) phenol: chloroform: isoamyl alcohol (25:24:1). Isopropanol was used for DNA precipitation, and Tris-EDTA (TE) buffer was used to dissolve the DNA total volume 30 mL/membranes set (0.45 μm + 0.025 μm). The DNA was stored at −20 °C for future metagenomic analysis. Total extracted DNA was confirmed by agarose gel electrophoresis.

### 2.4.2. DNA Extraction from Whole Blood Samples

DNA was extracted from whole blood samples (300 μL) using a Genomic DNA Isolation Kit (PDC11-0100; BIO-HELIX Co., Xindian Dist, New Taipei City, Taiwan). Briefly, 900 μL of buffer CR was added and mixed by inversion. The tubes were incubated at RT for 10 min and then centrifuged for 5 min at 4000× $g$. Next, 300 μL of buffer CC was added to the resuspended cells and incubated at 60 °C for 10 min until the sample lysates were clear. Then, 400 μL of buffer CB was added and shaken vigorously, followed by centrifugation at 12,000× $g$ for 1 min. Column CC was placed in a 2 mL collection tube. The clear supernatant from the previous step was transferred completely to the column and centrifuged at 14,000× $g$ for 30 s. The flow-through was discarded, 400 μL of the buffer W1 was added to the column CC, and samples were centrifuged at 14,000× $g$ for 30 s. The flow-through was discarded, and 600 μL of buffer W2 was added to column CC and centrifuged at 14,000× $g$ for 30 s. The flow-through was discarded, and the column was centrifuged at 14,000× $g$ for 2 min. Finally, buffer TE (50–200 μL) was added and centrifuged for 2 min at 14,000× $g$. The isolated DNA was then stored at −20 °C.

## 2.5. EBV DNA Detection by PCR

EBV DNA (*EBNA-2* gene) and an internal control (human *β-globin* gene) were amplified via PCR using *β-globin* forward (GH2O; 5′-GAAGAGCCAAGGACAGGTAC-3′) and reverse (PCO4; 5′-CAACTTCATCCACGTTCACC-3′) primers [63] and EBNA-2 forward (5′-CAGGTACATGCCAACAACCTT-3′) and reverse (5′-CCAACAAAGATTGTTAGTGG AAT-3′) primers [64]. The expected PCR product sizes were 268 bp and 219 bp for the *β-globin* and *EBNA-2* genes, respectively. The B95 cell line was used as a positive control. The PCR reaction was prepared using 5×FiREPOL ready-to-load master mix (Solis Bio Dyne, Tartu, Estonia) as follows: 1×FiREPOL master mix, 0.4 pM forward primer, 0.4 pM reverse primer, 3 μL DNA template, and distilled water (DW) to a 25 μL volume. PCR conditions were as follows: initial activation at 95 °C for 5 min, 40 cycles of denaturation at 95 °C for 1 min, annealing at 58 °C for 1 min, elongation at 72 °C for 1 min; and final elongation at 72 °C for 5 min. PCR products were visualized by 2% agarose gel electrophoresis in 1×Tris-acetate-EDTA buffer at 100 V for 40 min.

## 2.6. EBV DNA Detection by qPCR

EBV DNA (*EBNA-1* and *EBNA-2* genes) was also detected by qPCR using *EBNA-1* forward (QP3; 5′-CCACAATGTCGTCTTACACC-3′) and reverse (QP4; 5′-ATA ACAGA-CAATGGA CTCCCT-3′) primers. *EBNA-2* used the same PCR primers as in Section 2.5. The

expected PCR product for the EBNA-1 gene was 99 bp [30]. The B95 cell line was used as a positive control. The PCR reaction used 5×FiREPOL Eva Green qPCR Mix Plus (Solis Bio Dyne, Tartu, Estonia) prepared as follows: 1×FiREPOL master mix, 0.4 pM forward primer, 0.4 pM reverse primer, 2 μL DNA template, and DW up to a 20 μL volume. The qPCR conditions were as follows: initial activation at 95 °C for 12 min, 40 cycles of denaturation at 95 °C for 15 s, annealing/elongation at 60 °C for 30 s, and melting at 65–95 °C for 5 s/set.

### 2.7. Metagenomics

The DNA extracted from the tap water, natural water and the pooled human whole blood samples were analyzed by shotgun metagenomic sequencing. The extracted DNA represented all viruses in the water and whole blood samples. Fraction steps were checked using Agilent 2100 and qPCR to confirm that sufficient target DNA was present in the end repairing range. For tail construction, sequencing adapters were ligated, and PCR enrichment steps were used to create the library. Sequencing was performed using the Illumina platform after library clustering with paired-end reads.

### 2.8. Geographic Information System (GIS)

A GIS was used to connect the data on oncogenic virus prevalence in the Mueang District and show their distribution. GIS connected oncogenic virus data to a map and integrated the location data of 15 subdistricts, including Wiang, Mae Tam, Mae Na Ruea, Ban Tun, Ban Tam, Ban Tom, Mae Puem, Mae Ka, Ban Mai, Cham Pa Wai, Tha Wang Thong, Mae Sai, Ban Sang, Tha Champi, and San Pa Muang.

### 2.9. Statistical Analysis

The data were analyzed using the IBM SPSS software version 16, and $p < 0.05$ was considered statistically significant. Pearson's Chi-squared test was used to compare categorical variables between groups. An independent Student's *t*-test was used to compare separate mean ± standard deviation (SD) sets.

## 3. Results

### 3.1. Water Quality Analysis

Tap water sources from the Mae Ka (No. 1) and Wiang Subdistricts were collected from Mueang Phayao District, Phayao Province. Chemical and microbial analysis indicated that tap water from Mae Ka (No. 1) contained 0.330 mg/L Fe, 0.135 mg/L Mn, turbidity of 17.235 mg/L, and *E. coli* and total coliform bacteria numbers above the 2017 standard value from the Thai Provincial Waterworks Authority [65]. Chemical and microbial analysis of tap water from the Wiang Subdistrict showed that it also contained total coliform bacteria numbers above the standard value [65]. The results are shown in Table 1.

Chemical and microbial analysis of tap water from Mae Ka (No. 1) showed it contained higher than standard values, which are harmful to consumers. Its turbidity was visible to the naked eye, while tap water from the Wiang Subdistrict was clear. Tap water from both sources was contaminated with total coliform bacteria, which are harmful to the health of individuals in the community.

**Table 1.** The water turbidity, color, pH, and trace element analysis.

| | Property | Standard * | Mae Ka (No. 1) | Wiang | Unit |
|---|---|---|---|---|---|
| | Color | 15 | 0.480 | 0.700 | Pt.Co unit |
| Physical | Turbidity | 5 | 17.235 | 1.575 | NTU |
| | pH | 6.5–8.5 | 7.97 | 7.63 | |
| | Total dissolved solids | 1000 | 205.00 | 135.75 | mg/L |
| | Fe | 0.3 | 0.330 | 0.062 | mg/L |
| | Mn | 0.1 | 0.135 | 0.030 | mg/L |
| | Cu | 2.0 | 0.010 | 0.003 | mg/L |
| | Zn | 3.0 | 0.018 | 0.010 | mg/L |
| Chemical | Total Hardness ($CaCO_3$) | 300 | 99.5 | 71.0 | mg/L |
| | $SO_4^{2-}$ | | 6.525 | 12.695 | mg/L |
| | $Cl^-$ | 250 | Not detected | 16.715 | mg/L |
| | $F^-$ | 1.5 | 0.24 | 0.34 | mg/L |
| | $NO_3^-$ | 50 | 1.77 | Not detected | mg/L |
| | $NO_2^-$ | 3 | Not detected | Not detected | mg/L |
| Microbiological | Total coliform bacteria | - | 23.0 | 1.1 | MPN-100 mL |
| | *Escherichia coli* | - | Detected | Not detected | per 100 mL |
| | Hg | 1 | Not detected | Not detected | mg/L |
| Chemical Poison | Pb | 10 | 0.001 | Not detected | mg/L |
| | As | 10 | 0.002 | 0.001 | mg/L |
| | Cr | 50 | Not detected | Not detected | mg/L |
| | Cd | 3 | Not detected | Not detected | mg/L |

Note(s): Key: *, [65]; MPN, most probable number; NTU, nephelometric turbidity units.

### 3.2. Metagenomics Analysis

Extracted DNA from tap water, natural water, and pooled whole blood samples were analyzed by shotgun metagenomic sequencing. Surprisingly, the metagenomic analysis indicated the presence of pathogenic bacteria such as *E. coli*, *Salmonella enterica*, *Salmonella* spp., *Staphylococcus aureus*, *Legionella* spp., *Pseudomonas aeruginosa*, *Vibrio cholerae*, *Serratia marcescens*, *Klebsiella pneumoniae*, *Shigella sonnei*, *Shigella flexneri,* and *Burkholderia pseudomallei* in all the water samples, including tap water. The metagenomics analysis also revealed *Cryptosporidium* and *Giardia* spp. in all water samples, including tap water. Additionally, unidentified (virus groups) comprised 0.2% of all microbes.

In the pooled whole blood samples, *Ortervirales* (*Retroviridae*) comprised 72% of all viruses, *Herpesvirales* (*Herpesviridae*) 20%, *Caudovirales* (*Siphoviridae, Myoviridae*, and *Podoviridae*) 7%, and unidentified virus (*Mimiviridae, Iridoviridae, Poxviridae* (molluscum contagiosum virus), and *Phycodnaviridae*) 1%. In the water samples, *Ortervirales* (*Retroviridae*) comprised 0.02% of all viruses, *Herpesvirales* (*Herpesviridae*) 0.03–0.04% (7% in Wiang Subdistrict tap water), *Caudovirales* (*Siphoviridae, Myoviridae,* and *Podoviridae*) 79–84%, and unidentified viruses 0.1%. *Poxviridae* (molluscum contagiosum virus), a pathogenic DNA virus, was found in pooled human blood samples but not in water. However, the monkeypox virus was not found in either water or blood samples.

*Herpesvirales* was found in human blood and all water samples (Figure 1). *Equid gammaherpesvirus 5* was found in all water samples. *Herpesviridae* was the predominant oncogenic DNA virus, such as oncogenic HHV-8, in both natural water sources. *Herpesviridae* was the predominant disease-causing DNA virus found in human blood and water samples. In pooled human blood samples, EBV comprised 2% of total *Herpesviridae*. In addition, EBV comprised 54% of all *Herpesviridae* in Mae Ka (No. 1) tap water, 89% in Wiang subdistrict tap water, 43% in the Huai Na Poi water, and 37% in the Kwan Phayao water. Tap water from Mae Ka (No. 1) and Wiang Subdistrict, natural water from Huai Na Poi and Kwan Phayao, and pooled human blood samples contained 6.42, 100.12, 6.36, 7.66 EBV copies/L, and 4.53 EBV copies/100 blood samples. This study shows that EBV is found in tap and natural water sources and human blood samples. *Human betaherpesvirus 6B* was found

in the natural water from Kwan Phayao, *Human betaherpesvirus 6A* was found in Mae Ka (No. 1) tap water and human blood sample.

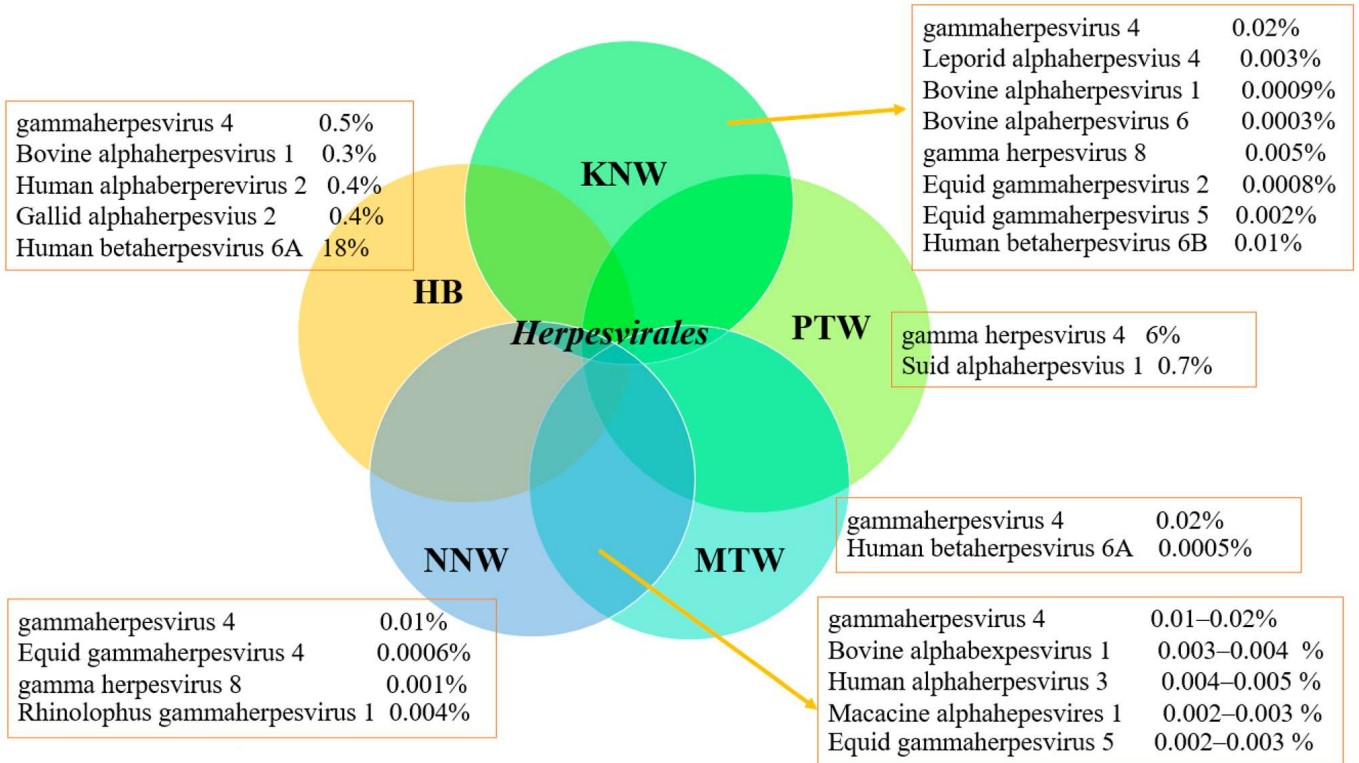

**Figure 1.** *Herpesvirales* in human blood and all water samples. Percentage (%) values represent the percentage of the virus among all viruses. Key: HB, pooled human blood samples; MTW, tap water from Mae Ka (No. 1); PTW, tap water from Wiang Subdistrict; KNW, natural water from Kwan Phayao; NNW, natural water from Huai Na Poi.

The sequences of EBV from natural and tap water were similar to EBV reference strain (RefSeq) NC_007605 (171,823 bp DNA circular, *Human gammaherpesvirus 4*, complete genome, version NC_007605.1), mostly at location about 800–102 kb. The sequence of EBV from human blood was similar to EBV reference strain (RefSeq) NC_007605 (171,823 bp DNA circular, *Human gammaherpesvirus 4*, complete genome, version NC_007605.1) at location about 800–102 kb but not the same location with water. Therefore, in all samples (human blood, tap water, and natural water) were found *Human gammaherpesvirus 4* (such as at the location of *EBNA-1* gene) with mostly percentage identity up to 100% when using cutoff >50 bp. However, when using a <50 bp cutoff, the result found that human blood samples and all types of water were similar with EBV reference strain (RefSeq) NC_007605 at the same location with percentage identity mostly up to 100% (Figure 2, Supplementary Figures S1 and S2).

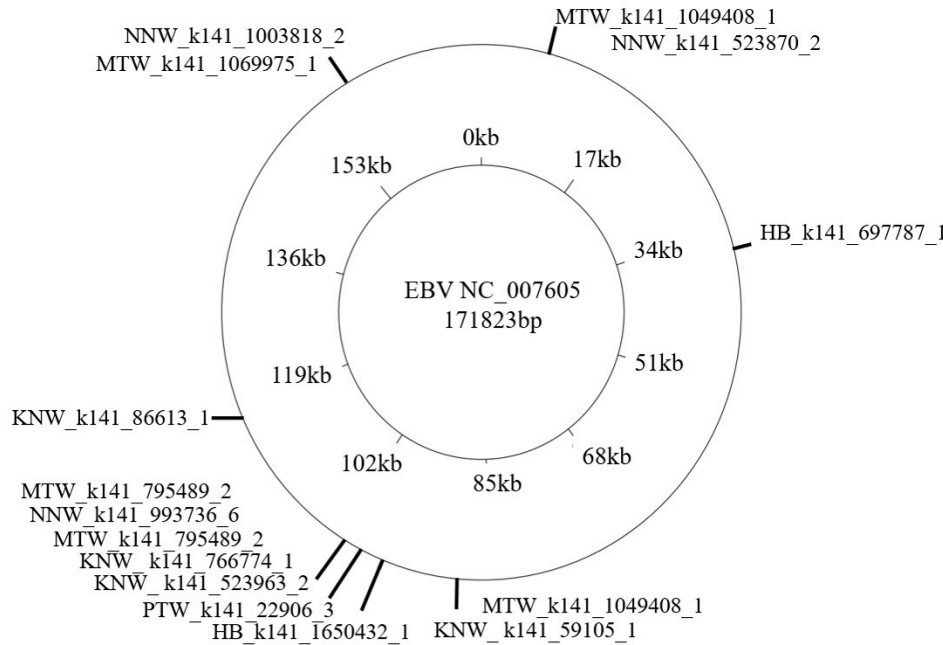

**Figure 2.** The examples sequence location of EBV from blood and water on EBV RefSeq NC_007605 (171,823 bp complete genome) when using cutoff >50 bp. EBV were mostly found at the location about 800 kb–102 kb. Key: HB, pooled human blood samples; MTW, tap water from Mae Ka (No. 1); PTW, tap water from Wiang Subdistrict; KNW, natural water from Kwan Phayao; NNW, natural water from Huai Na Poi.

### 3.3. EBV DNA Detection by PCR and qPCR and Environment Factors

Human blood samples were collected from 813 blood donors in Phayao Province, comprising 231 men (28.41%) and 582 women (71.59%) aged 3–90 years (mean 52.01 ± 18.65 years). PCR identified 7/813 (0.86%) EBV-positive samples (*EBNA-2* gene), while qPCR identified 6/813 (0.74%) and 53/813 (6.52%) samples based on *EBNA-2* and *EBNA-1*, respectively. EBV prevalence based on all three methods was 59/813 (7.26%; Table 2). EBV prevalence differed significantly by method ($p > 0.05$). *EBNA-1* gene detection by qPCR showed higher sensitivity in blood than *EBNA-2* gene detection. *EBNA-2* gene detection was also inconsistent between PCR and qPCR.

**Table 2.** Life history and EBV status (%).

| Demographical Factor | All Three Methods (%) | | *p*-Value |
| --- | --- | --- | --- |
| | Positive | Negative | |
| **Sex** | | | |
| Male | 17 (7.36) | 214 (92.64) | 0.944 |
| Female | 42 (7.22) | 540 (92.78) | |
| **Age (years)** | | | |
| Mean | 44.37 | 52.61 | 0.000 |
| SD | 16.05 | 18.71 | |
| **Age Groups (years)** | | | |
| 1–10 | 0 (0) | 21 (100) | |
| 11–20 | 7 (12.28) | 50 (87.72) | |
| 21–30 | 6 (11.54) | 46 (88.46) | |
| 31–40 | 5 (9.62) | 47 (90.38) | |
| 41–50 | 20 (16.81) | 99 (83.19) | |
| 51–60 | 15 (7.54) | 184 (92.46) | |
| 61–70 | 2 (0.93) | 214 (99.07) | |
| 71–80 | 4 (5.19) | 73 (94.81) | |
| 81–90 | 0 (0) | 20 (100) | |

**Table 2.** *Cont.*

| Demographical Factor | All Three Methods (%) | | *p*-Value |
|---|---|---|---|
| | Positive | Negative | |
| **Congenital Disease** | | | |
| Yes | 23 (5.96) | 363 (94.04) | 0.175 |
| No | 36 (8.43) | 391 (91.57) | |
| **Family History of Cancer** | | | |
| Yes | 16 (8.70) | 168 (91.30) | 0.392 |
| No | 43 (6.84) | 586 (93.16) | |
| **Exercise** | | | |
| Yes | 48 (7.35) | 605 (92.65) | 0.835 |
| No | 11 (6.87) | 149 (93.13) | |
| **Boiled/Filtered Drinking Water** | | | |
| Yes | 41 (7.65) | 495 (92.35) | 0.549 |
| No | 18 (6.50) | 259 (93.50) | |
| **Bottled Drinking Water** | | | |
| No | 8 (12.70) | 55 (87.30) | 0.083 |
| Yes | 51 (6.80) | 699 (93.20) | |
| **Cleaning of Water for Consumption** | | | |
| Yes | 46 (6.73) | 638 (93.27) | 0.178 |
| No | 13 (10.08) | 116 (89.92) | |
| **Tap Water Used for Brushing Teeth** | | | |
| No [a] | 14 (12.17) | 101 (87.83) | 0.028 |
| Yes | 45 (6.45) | 653 (93.55) | |
| **Alcohol Consumption** | | | |
| Yes | 35 (11.40) | 272 (88.60) | 0.000 |
| No | 24 (4.74) | 482 (95.26) | |
| **Serving Spoon** | | | |
| No | 13 (10.16) | 115 (89.84) | 0.168 |
| Yes | 46 (6.72) | 639 (93.28) | |
| **Smoking Status** | | | |
| Yes | 13 (10.74) | 108 (89.26) | 0.109 |
| No | 46 (6.65) | 646 (93.35) | |
| **Secondhand Smoke Status** | | | |
| Yes | 20 (11.76) | 150 (88.24) | 0.011 |
| No | 39 (6.07) | 604 (93.93) | |
| **Eat Fresh Fruit (Per Week)** | | | |
| No | 6 (19.35) | 25 (80.65) | 0.019 |
| 1–2 times | 8 (9.64) | 75 (90.36) | |
| 3–4 times | 17 (8.50) | 183 (91.50) | |
| 5–7 times | 28 (5.61) | 471 (94.39) | |
| **Eat Vegetables (per Week)** | | | |
| No | 3 (8.57) | 32 (91.43) | 0.236 |
| 1–2 times | 6 (10.17) | 53 (89.83) | |
| 3–4 times | 15 (10.49) | 128 (89.51) | |
| 5–7 times | 35 (6.08) | 541 (93.92) | |

Note(s): The three methods were *EBNA-2* PCR, *EBNA-1* qPCR, and *EBNA-2* qPCR. Key: [a], used other water, such as rain/artesian water, for brushing teeth.

Age, congenital disease, family cancer history, alcohol consumption, secondhand smoke, drinking water source (bottled water), and tap water used for brushing teeth and eating fresh fruit were associated with EBV DNA positivity in blood, depending on the detection method used. BMI, exercise, smoking, drinking water source (boiled or filtered), cleaning of water for consumption, and water used with vegetables were not significantly associated with EBV DNA detection, depending on the detection method used. The results are shown in Table 2 and Figure 3.

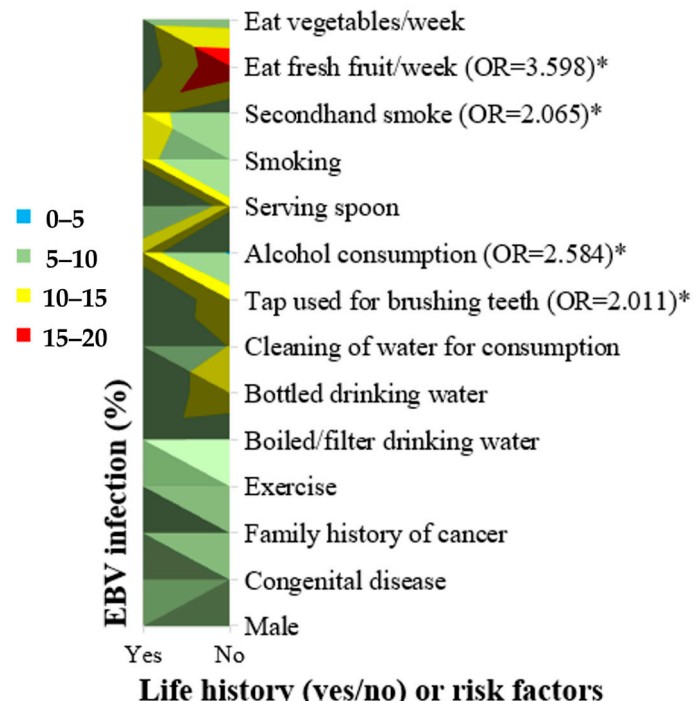

**Figure 3.** Life history (yes/no) and risk factors for EBV infection (%) in blood based on the three methods. The color represents the percentage of exposure to environmental risk factors: 0–5%, blue; 5–10%, green; 10–15%, yellow; and 15–20%, red. Male, congenital disease, family cancer history, exercise, and boiled/filtered drinking water were not risk factors for EBV DNA detection in blood. Not using tap water for brushing teeth, alcohol consumption, secondhand smoke exposure, and not eating fresh fruit weekly were risk factor factors for EBV DNA detection in blood. Note: not using tap water for brushing teeth reflects the use of other water sources, such as rain/artesian water. Key: *, statistically significant ($p < 0.05$).

Individuals with congenital disease had higher PCR-based *EBNA-2* gene positivity (6/386 (1.55%)) than those with no congenital disease (1/427 (0.23%); $p = 0.042$). Individuals with a family cancer history had higher qPCR-based *EBNA-2* gene positivity (4/184 (2.17%)) than those with no family cancer history (2/629 (0.32%); $p = 0.010$; odds ratio = 6.967; 95% CI: 1.266–38.344). Individuals who did not use bottled water had a higher qPCR-based *EBNA-1* gene positivity (8/63 (12.69%)) than those who used bottled drinking water (45/750 (6.00%); $p = 0.039$; odds ratio = 2.279; 95% CI: 1.023–5.074).

Combining the EBV-positive results across the three methods, 59/813 (7.26%) samples were EBV positivity: 17/231 (7.36%) in men and 42/582 (7.22%) in women ($p = 0.944$). In addition, many factors were associated with EBV DNA infection in blood samples. The mean age of EBV-positive individuals was 44.37 ± 16.05 years, lower than EBV-negative individuals (52.61 ± 18.71 years; $p = 0.000$). No EBV DNA was detected in blood samples from children aged 3–10 years and individuals aged 81–90 years. The 41–50 age group showed higher EBV positivity (20/119 (16.81%)) than the 11–20 (7/57 (12.28%)), 21–30 (6/52 (11.54%)), 31–40 (5/52 (9.62%)), 51–60 (15/199 (7.54%)), 71–80 (4/77 (5.19%)), and 61–70 (2/216 (0.93%)) age groups ($p = 0.000$).

Individuals who used other water sources, such as rain/artesian water, for brushing teeth had a higher EBV positivity (14/115 (12.17%)) than those who used tap water (45/698 (6.45%); $p = 0.028$; odds ratio = 2.011; 95% CI: 1.066–3.797). Individuals who consumed alcohol had a higher EBV positivity (35/307 (11.40%)) than those who did not (24/506 (4.74%); $p = 0.000$; odds ratio = 2.584; 95% CI: 1.506–4.436). Individuals exposed to secondhand smoke had a higher EBV positivity (20/170 (11.76%)) than those not exposed to secondhand smoke (39/643 (6.07%); $p = 0.011$; odds ratio = 2.065; 95% CI: 1.170–3.644). Individuals who did not eat fresh fruit had a higher EBV positivity (6/31 (19.35%)) than

those who ate fresh fruit 1–7 times per week (53/782 (6.78%); *p* = 0.019; odds ratio = 3.598; 95% CI: 1.404–9.218). The results are presented in Table 2, and the odds ratios are shown in Figure 3.

*3.4. GIS*

To assess the epidemic spread of EBV in various sub-districts in Phayao province, Thailand, GIS was used to connect and integrate EBV prevalence data from human blood samples across 15 subdistricts in the Mueang District and show their distribution as a map (Figure 4).

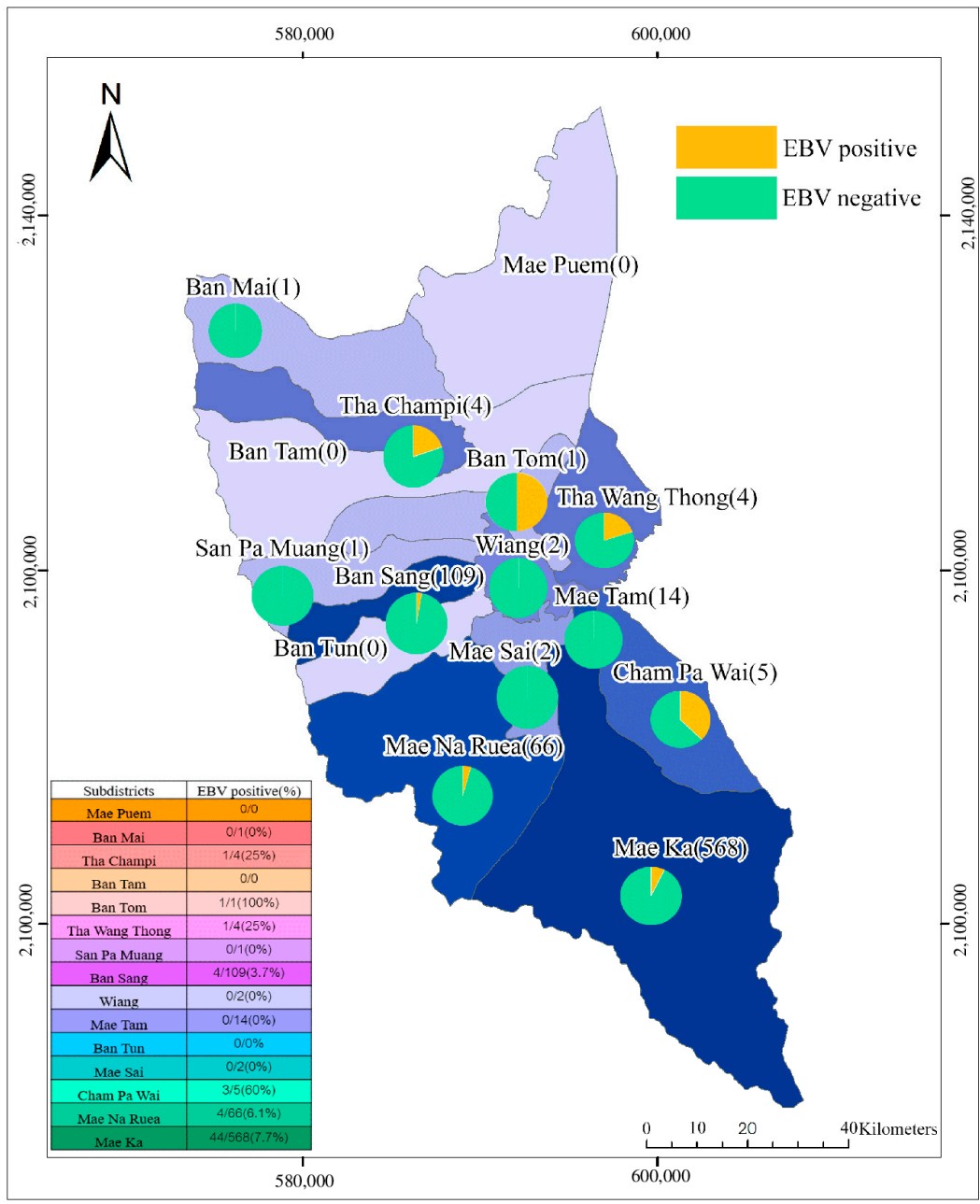

**Figure 4.** GIS-connected oncogenic EBV prevalence data from human blood samples in the 15 Mueang subdistricts highlights the EBV distribution. EBV-positive means EBV-positive in human blood.

## 4. Discussion

The 2017 report by the Thai Provincial Waterworks Authority [65] indicated that total coliform bacteria, *S. aureus, Salmonella* spp., *Clostridium perfringens, E. coli,* and *Shigella* spp. were not detected. However, this study detected all these pathogens in both tap water sources. Therefore, tap water quality is not passing the standards of the Provincial Waterworks Authority, affecting the health of the residents in the district.

The Thai Metropolitan Waterworks Authority has increased the testing requirements for viruses to determine the safety of tap water [66]. This study did not address RNA viruses. However, it detected pathogens in both tap water sources that are prohibited by these standards, including total coliform bacteria, *S. aureus, Salmonella* spp., *E. coli, Shigella* spp., *Legionella* spp., *P. aeruginosa, V. cholerae, C. perfringens, Cryptosporidium* spp., and *Giardia* spp. Producing tap water from dirty water sources can cause many problems, and its safety should be taken seriously. Even though the tap water sources from Phayao are unsuitable for drinking, most of the population uses tap water to brush their teeth, wash vegetables and fruit, or wash dishes, which may be a channel for bacterial and viral diseases. In particular, the oncogenic viruses found in this study (EBV) were present in large quantities in water sources.

This study found that the sequence of oncogenic EBV from natural water, tap water, and human blood sample was similar to the EBV reference strain, *Human gammaherpesvirus 4*, complete genome (version NC_007605.1). Possibly, there are associations between oncogenic EBV in water and humans and further study should be conducted on the risk of oncogenic EBV in water being transmitted to humans.

Herpesviruses (equine herpesvirus type-1) have been found to persist in various water environments for up to three weeks in controlled experiments [67]. This study found equid gammaherpesvirus 5 in all water sources, which causes chronic lung disease and equine multinodular pulmonary fibrosis in adult horses [68]. The herpes simplex virus (HSV) survived 4 h in tap water, 24 h in distilled water, and 4.5 h on plastic surfaces, suggesting that fomites such as these may be nonvenereal routes for HSV transmission in spa water [69]. Therefore, oncogenic EBV in water suggests that transmission to humans is possible and should be investigated further.

This study is the first to discuss the safety of natural water, tap water and community health in Thailand in the EBV context. Currently, non-communicable diseases account for 71% of deaths in Thailand, followed by cancer (17%) [70]. Many factors, including viruses, cause these diseases. Water is frequently a vehicle for viral transmission. Virus survival is higher in sterile water [11–13]. This study provides an overview of viral communities that exist in natural water sources in the large Phayao District using metagenomics. Therefore, it can serve as a future guideline for public health agencies and the Provincial Waterworks Authority.

This study found that individuals aged 11–50 were most likely to be infected due to the increased frequency of close social contact with infected individuals. Additionally, EBV infections might play a role in individual immune responses, which are more severe in adolescence or adulthood [38]. EBV prevalence differed based on the detection method used. Therefore, several methods or genes are required for EBV detection. This study found the low prevalence of blood-borne EBV infection. Thailand is in a tropical zone and has year-round sunshine. Sunlight and vitamin D are believed to protect against autoimmunity by increasing the number of CD8+ T cells available to control low-level EBV infections [71].

EBV transits between epithelial and B cells. The first innate barrier to infection is the oropharyngeal epithelium. EBV replicates mainly in epithelial cells in the oropharynx. Therefore, EBV is believed to undergo replication to be efficiently transmitted and spread to other hosts via saliva [72]. Contact through saliva, such as kissing, sharing glasses, and not using a serving spoon or utensils, may cause EBV infection in epithelial cells but not in blood. Therefore, this study did not find an association between these factors and blood-borne EBV infection. EBV spreading from epithelial cells to the blood might be associated with other factors, such as host immune responses. EBV DNA in whole

blood is a superior prognostic and monitoring factor in many diseases, such as diffuse large B-cell lymphoma [46]. Therefore, the presence of blood-borne EBV may be clinically significant [38] and for blood transfusion.

This study found fewer blood-borne EBV infections in individuals who used tap water for brushing their teeth compared to rain or artesian water, which is still used in rural areas. The Mueang District is the center of Phayao Province, which is a small province, and the authority does not provide tap water. In some areas, villages still produce tap water, and rain and artesian water are still used. Therefore, water quality might not be appropriately controlled, and contact with infectious agents via water may be a concern based on the water source used.

The *EBNA-1* gene is important for cell immortalization [73], and the *EBNA-2* gene is essential for establishing a latent EBV infection and B-cell immortalization [74,75]. Individuals not using bottled water had a higher *EBNA-1* gene positivity than those who used bottled water. While bottled water is generally considered safe, its sterilization process remains of concern [76,77].

This study found that environmental factors such as age, alcohol consumption, secondhand smoke exposure, and not eating fresh fruit were risk factors for blood-borne EBV infection. Smoking was associated with EBV infection [78] and increased risk of EBV-positive Hodgkin lymphoma (odds ratio = 1.4; 95% CI: 1.1–1.9) [79], gastric cancer [80], and NPC [81]. However, no mention of secondhand smoke exposure was found in previous studies. In addition, they found no significant association between EBV and alcohol consumption in NPC [80,81] or gastric cancer [80]. However, this study found that blood-borne EBV infection was associated with alcohol consumption. Nevertheless, several studies have found an association between habitual alcohol consumption and NPC and OSCC risk [82–85].

This study found that eating fresh fruit may be protective against NPC caused by blood-borne EBV infection [86,87]. This study also found that BMI was not associated with blood-borne EBV infection, different from Keegan et al. [88]. While obesity has been shown to impact the host immune response to infections that can induce inflammation, its association remains unclear [38]. A family cancer history has been previously associated with NPC [89], according to this study.

## 5. Conclusions

The findings of this study support the conclusion that water in developing countries continues to be unsafe. Human oncogenic EBV was present in natural and tap water. Finally, basic knowledge about tap water sterilization processes and viral control should be promoted. Blood-borne oncogenic EBV should be a concern for blood transfusion in patients. Therefore, the detection of oncogenic EBV in water suggests that transmission via water is possible and should be investigated further.

**Supplementary Materials:** The following are available online at https://www.mdpi.com/article/10.3390/w15020323/s1, Supplementary File S1: metagenomic analysis using a cutoff of more than 50 bp and Supplementary File S2: metagenomic analysis using a cutoff of less than 50 bp. Supplementary Figures S1 (The sequence location of EBV from blood and water on EBV) and S2 (The alignment sequence of EBV from water compare with EBV) in Supplementary File S3.

**Author Contributions:** Conceptualization, S.B. (Sureewan Bumrungthai); methodology, S.B. (Sureewan Bumrungthai), S.P. (Sutida Pongpakdeesakul) and N.I.; validation, S.B. (Sureewan Bumrungthai) and S.P. (Sutida Pongpakdeesakul); investigation, S.B. (Sureewan Bumrungthai); data curation, S.B. (Sureewan Bumrungthai) and S.P. (Sutida Pongpakdeesakul); writing—original draft preparation, S.B. (Sureewan Bumrungthai) and S.P. (Sutida Pongpakdeesakul); writing—review and editing, C.P., T.E. and S.B. (Sureewan Bumrungthai); funding acquisition, S.B. (Sureewan Bumrungthai); investigation, S.B. (Sureewan Bumrungthai), T.E., C.P., S.B. (Surachat Buddhisa), K.M., A.P., W.S., S.P. (Supaporn Passorn), P.C. and S.D. All authors have read and agreed to the published version of the manuscript.

**Funding:** This research was funded by the University of Phayao (grant no. FF64-RIB005).

**Institutional Review Board Statement:** This study was approved by the Committee on Human Research Ethics in Health Sciences and Science and Technology, University of Phayao (Mae Ka, Thailand; 1.3/023/63). All procedures involving human participants performed in the study were in accordance with the ethical standards of the Declaration of Helsinki, the Belmont Report, the Council for International Organizations of Medical Sciences guidelines, and the International Conference on Harmonization in Good Clinical Practice.

**Informed Consent Statement:** Informed consent was obtained from all subjects involved in the study.

**Acknowledgments:** We thank the Thalassemia Unit, University of Phayao for use of the real-time PCR machine. Chamnan Sangkeo and the Adventure team, University of Phayao. Ati Burussakarn, and Watcharapong Panthong from Khon Kean University for preparation for the control of EBV.

**Conflicts of Interest:** The authors declare no conflict of interest.

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
