# Peer review of "Human Oncogenic Epstein–Barr Virus in Water and Human Blood Infection of Communities in Phayao Province, Thailand"

_water, doi:10.3390/w15020323_

Round 1

Reviewer 1 Report (Previous Reviewer 1)

The authors have undertaken important investigations on the presence of EBV virus in water. What I still miss here is clear description of the connection between presence of EBV in water and the presence of EBV in blood.

Fig. 6 - What is shown here? There is only 1-sentence comment and no description. What does "EBV positive"mean  here? Do you mean people or water samples? And what is the correlation of EBV presence between water samples and human blood samples in a given area?

Author Response

Dear reviewer 

Reviewer 2 Report (New Reviewer)

I think this study is interesting in that it provides data on the occurrence of EBV in water in a developing county which has not received a lot of attention. Since EBV can be spread by salvia it seems possible that it could be transmitted by untreated drinking water.  

I think the sentence in the abstract and conclusions “Therefore, it is important to highlight that water transmission to human of oncogenic EBV might be further study.” Should be restated as “The detection of EBV in water suggests that transmission via water is possible and should be investigated further.”

More details on the method of virus concentration need to be provided. Also it is stated that 5 liters of water was passed through a 47 mm diameter filter with a 0.025 µm porosity. Having filtered many water samples in my career I do not see how this is possible – I would clog long before 5 liters was passed through it – even treated tap water getting a liter through that size filter would not be possible. Also details on how the virus was “extracted” from the filters needs to be provided. What was the volume of the extract? Was it concentrated any further. A lot more details are needed.

Details on how the E. coli and coliforms were detected or not need to be included.

I think there is too much background on EBV that could be detected as it is not needed.

Author Response

Dear reviewer 

This manuscript is a resubmission of an earlier submission. The following is a list of the peer review reports and author responses from that submission.

Round 1

Reviewer 1 Report

The paper "Epstein-Barr virus (EBV) in tap water. The risk of oncogenic EBV infection in human blood" describes identification of EBV virus in tap water in one of the districts in Thailand.

The text is not written properly, the description and presentation of the results is very poor, there is a lot of mess. For example Fig. 2 - is it EBV identified in water or  in the blood of the inhabitants?

Table 2 and 3 are too big, hard to understand what is presented. data from these table should be presented in different way, more clear.

How many virus copies was found in tap water? You wrote only "large quantities" what is not scientific description.

The authors wrote in the introduction that EBV genes were found in 61% od healthy blood donors. If its prevalence is so high, how do you know that it is caused by tap water drinking? Additional experiments are required to make such conclusion.

Were similar studies made in tap water in other regions?

The conclusions are not written well, the results of the studies should be clearly stated.

The text need deep linguistic corrections.

Reviewer 2 Report

There are several technical and design issues. For example, it is stated that the water samples were filtered with a vacuum pump using MF-Millipore MCE membranes with 0.025 µm pores and it appears that the DNA extractions are done on what is retained on the filter membranes. However, viruses are not effectively retained by these membranes; therefore, the resulting quantification is largely biased.

In the results section, it appears that a very small number of people are infected with EBV. This statement is very surprising because EBV is a ubiquitous virus (more than 95% of the adult population is infected). The technique used to detect the presence of EBV is obviously not adapted, it would be good to do a serological test which would certainly change the results. It is likely that the measurement taken does not reflect the real infection.

The authors claim without scientific argument that EBV in the water (it is not known whether it is infectious) is responsible for the infection of the people. There are no experimental results to support this. Their conclusion is therefore not scientific.